# Unraveling the Pathogenetic Mechanisms Underlying the Association between Specific Mitochondrial DNA Haplogroups and Parkinson’s Disease

**DOI:** 10.3390/cells13080694

**Published:** 2024-04-17

**Authors:** Min-Yu Lan, Tsu-Kung Lin, Baiba Lace, Algirdas Utkus, Birute Burnyte, Kristina Grigalioniene, Yu-Han Lin, Inna Inashkina, Chia-Wei Liou

**Affiliations:** 1Department of Neurology, Kaohsiung Chang Gung Memorial Hospital and Chang Gung University College of Medicine, Kaohsiung 83301, Taiwan; myl@ksts.seed.net.tw (M.-Y.L.); tklin@adm.cgmh.org.tw (T.-K.L.); 2Center for Mitochondrial Research and Medicine, Kaohsiung Chang Gung Memorial Hospital and Chang Gung University College of Medicine, Kaohsiung 83301, Taiwan; cgmhlinyh202@cgmh.org.tw; 3Riga East Clinical University Hospital, Latvia Institute of Clinical and Preventive Medicine, University of Latvia, LV-1038 Riga, Latvia; 4Department of Human and Medical Genetics, Institute of Biomedical Sciences, Faculty of Medicine, Vilnius University, 03101 Vilnius, Lithuania; algirdas.utkus@mf.vu.lt (A.U.); birute.burnyte@santa.lt (B.B.); kristina.grigalioniene@santa.lt (K.G.); 5Latvian Biomedical Research and Study Center, LV-1067 Riga, Latvia

**Keywords:** Parkinson’s disease, mitochondrial haplogroup, cybrid, transcriptome, unfolding protein response

## Abstract

Variants of mitochondrial DNA (mtDNA) have been identified as risk factors for the development of Parkinson’s disease (PD). However, the underlying pathogenetic mechanisms remain unclear. Cybrid models carrying various genotypes of mtDNA variants were tested for resistance to PD-simulating MPP^+^ treatment. The most resistant line was selected for transcriptome profiling, revealing specific genes potentially influencing the resistant characteristic. We then conducted protein validation and molecular biological studies to validate the related pathways as the influential factor. Cybrids carrying the W3 mtDNA haplogroup demonstrated the most resistance to the MPP^+^ treatment. In the transcriptome study, *PPP1R15A* was identified, while further study noted elevated expressions of the coding protein GADD34 across all cybrids. In the study of GADD34-related mitochondrial unfolding protein response (mtUPR), we found that canonical mtUPR, launched by the phosphate eIF2a, is involved in the resistant characteristic of specific mtDNA to MPP^+^ treatment. Our study suggests that a lower expression of GADD34 in the late phase of mtUPR may prolong the mtUPR process, thereby benefitting protein homeostasis and facilitating cellular resistance to PD development. We herein demonstrate that GADD34 plays an important role in PD development and should be further investigated as a target for the development of therapies for PD.

## 1. Introduction

Parkinson’s disease (PD) is characterized by dopaminergic neuron degeneration in the substantia nigra pars compacta as well as Lewy body accumulations in various brain regions, which are primarily composed of misfolded alpha-synuclein. The protein aggregation and subsequent loss of neurons lead to neurotransmission disruptions, particularly within the basal ganglia circuitry, leading to the motor and non-motor symptoms observed in PD patients [1]. Additionally, mitochondrial-derived complex I respiratory chain enzyme deficiency in the substantia nigra and inhibition of this enzyme can cause nigrostriatal degeneration and parkinsonism, highlighting the role of mitochondria in the pathogenesis of PD [2,3]. Indeed, several studies have demonstrated associations between mitochondrial dysfunction and PD [4]. Furthermore, variants of mitochondrial DNA (mtDNA), which inherently affect mitochondrial energy production and free radical formation, have been linked to PD development [5].

Associations between mtDNA variant-determined haplogroups and either resistance or vulnerability to Parkinson’s disease (PD) have previously been reported among various population groups [6,7]. These associations have been further supported by a meta-analysis of ethnic Caucasian population groups and cellular studies using cybrids harboring common Asian mtDNA haplogroups [8,9]. Meanwhile, the different biological characteristics noted in cybrids harboring the same nucleus but different mtDNA indicate that the discrepancies are due to either the influence of mtDNA on mitochondrial function or functional disparities in the communication pathways between the mtDNA and nuclear genome [10,11]. Although the exact pathomechanism underlying the association between mtDNA variants and PD remains unclear, several studies of familial PD cases have identified the involvement of genes and their related coding proteins acting within mitochondria to maintain protein homeostasis [12].

To monitor and regulate protein homeostasis for maintaining cellular survival, the unfolded protein response (UPR) is a critical intracellular signaling pathway [13]. Mitochondrial UPR (mtUPR) serves as the quality control machinery for imported and self-translated proteins through mitochondria-nuclear communication, which modulates the transcription of mitochondrion-specific protein folding helpers. Due to the relatively high oxidative stress burden on the proteins within the mitochondrial compartment, the proteins are prone to dysregulation. This process may be accelerated under various pathological insults, thereby causing mitochondrial proteins to misfold and activating the molecular cascades involved in mtUPR, which is triggered to recover protein homeostasis by modulating transcriptions of nucleus-encoded chaperones and proteases [14]. Indeed, mtUPR dysfunction and the subsequent failure of mitochondrial proteostasis have been implicated in PD development [15]. In our previous report comparing 725 PD patients and 744 non-PD controls, we revealed a lower odds ratio (0.50, 95% CI: 0.32–0.78) for PD development among Taiwanese people carrying the B5 mitochondrial haplogroup [9]. In the present study, by using a cytoplasmic hybrid (cybrid) cell model carrying various subtypes of mtDNA haplogroups encompassing the Asian and Caucasian populations, we further investigated gene expression alterations related to resistance to a PD-inducing neurotoxin. We found that mtUPR, specifically the temporal changes in the canonical pathway activity, may underlie the resistance of PD-protective mitochondrial haplogroups.

## 2. Materials and Methods

### 2.1. Cybrid Construction and Culturing

We used platelets from two groups of healthy volunteers of both ethnic Chinese (3 male and 4 female) and Caucasian (3 male and 3 female) backgrounds, ages ranging from 20 to 30, as mtDNA donors for the creation of cybrid cell lines. A total of thirteen platelet samples, each containing different mtDNA haplogroups, were fused with mitochondria-depleted osteosarcoma cells (143B rho), as described in our previous publication [16]. Using platelets as the source of mtDNA and mitochondria-depleted osteosarcoma cells as the host cells has been previously shown to be an effective method. Additionally, as platelet isolation is noninvasive in nature, platelet mitochondria present prolonged viability, and the mitochondrial-transfer procedure is relatively simple and efficient, we employed this method for this investigation [17]. The resulting cybrid cells created from the ethnic Chinese group contained haplogroups B4, B5, D4, D5, F1, F2, and N9a. The cybrids of the Caucasian group contained haplogroups H11, I3, J1, U5, V7, and W3. These cybrid lines were developed under the support of a cooperative project involving Taiwan, Latvia, and Lithuania. The study was performed in accordance with the Declaration of Helsinki. Written informed consent was obtained from all participants in accordance with protocols approved by the Institutional Review Boards of Kaohsiung Chang Gung Memorial Hospital of Taiwan (IRB No. 103-4459B, approved 20 January 2015). Protocols were also approved by the Central Committee of Medical Ethics of Latvia (N27 approved from 14 December 2011; and No. 2016-5, chapter 2, approved 24 November 2016) and the Vilnius Regional Biomedical Research Ethics Committee of Lithuania (No 158200-15-771-288, approved 3 February 2015).

### 2.2. Cellular Studies

#### 2.2.1. PD Simulating Model

To mimic a neurotoxic condition, we treated cybrids with different concentrations of MPP^+^ (1-methyl-4-phenylpyridinium). MPP^+^ is an environmental toxin known to be an inhibitor of the mitochondrial respiratory complex I enzyme and has been demonstrated to cause symptoms mimicking PD in humans [18]. Growth conditions were subsequently conducted and recorded.

#### 2.2.2. Cellular Viability Assay

All thirteen fused cybrid lines were separated into their respective Asian and Caucasian groups and then tested for their viability under various concentrations of MPP^+^ treatment. Cybrid cells were used within a narrow range of passage numbers (approximately 10 passages), and cell samples were collected on different days for each experiment. The cytotoxic effects on B4, B5, D4, D5, F1, F2, and N9 cells were assessed by WST-1 assay (Roche Diagnostics GmbH, Mannheim, Germany), while H, I, J, T, U, V, and W3 cells were assessed by both Alamar Blue (BioRad, Hercules, CA, USA) and WST-1 assay. After exposure to MPP+ (1, 2, 3, and 4 mM) for 24 h, 10% (*v*/*v*) of Alamar Blue solution or WST-1 reagent was added to each well, and the cells were incubated in the dark for an additional 1.5 (Alamar Blue, Bio-Rad, Hercules, CA, USA) or 2 h (WST-1, Roche Diagnostics GmbH, Mannheim, Germany) at 37 °C. The fluorescence for Alamar Blue (excitation at 560 nm and emission at 590 nm) or absorbance for WST-1 (450/630 nm) was measured by a multiplate reader. All experiments were performed in triplicate. Results were expressed as a percentage of control.

### 2.3. RNA Sequencing and Transcriptome Analysis

#### 2.3.1. RNA Isolation

To minimize sample variation, total RNA samples from three independent samples per treatment condition were pooled and then sent for whole-genome RNA next-generation sequencing (RNA-Seq), performed by Welgene Biotech Co., Ltd. (Taipei, Taiwan). Total RNA was extracted by Trizol^®^ Reagent (Invitrogen, Carlsbad, CA, USA) according to the instruction manual. Purified RNA was quantified at OD260 nm by using an ND-1000 spectrophotometer (Nanodrop Technology, Wilmington, DE, USA) and qualified by using a Bioanalyzer 2100 (Agilent Technology, Santa Clara, CA, USA) with RNA 6000 labchip kit (Agilent Technologies, Santa Clara, CA, USA).

#### 2.3.2. Library Preparation and Sequencing

All procedures were carried out according to the manufacturer’s protocol from Illumina (Illumina Inc., San Diego, CA, USA). Library construction of all samples was used by Agilent’s SureSelect Strand-Specific RNA Library Preparation Kit for 75 bp (single-end) sequencing on the Illumina NextSeq 500 platform. The sequence was directly determined using sequencing-by-synthesis technology via the TruSeq SBS Kit. Raw sequences were obtained from the Illumina Pipeline software bcl2fastq v2.0 and expected to generate 20 million reads per sample.

#### 2.3.3. RNA-Sequencing Analysis

Initially, the sequences generated were filtered to obtain qualified reads. Trimmomatics was implemented to trim or remove the reads according to the quality score. Qualified reads after filtering low-quality data were analyzed using TopHat/Cufflinks for gene expression estimation. The gene expression level was calculated as FPKM (fragments per kilobase of transcript per million mapped reads). For differential expression analysis, CummeRbund was employed to perform statistical analyses of gene expression profiles. The reference genome and gene annotations were retrieved from the Ensemble database.

### 2.4. Consistency of RNA Findings in Cell Lines and Protein Study for Pathogenesis

#### 2.4.1. RNA Reverse Transcription

Extraction of total RNA from cells was performed using TRIzol Reagent (Invitrogen) according to the manufacturer’s instructions. A total of 500 ng of total RNA was reverse-transcribed using the PrimeScript™ RT reagent Kit (Takara Bio, Inc., Otsu, Japan). The resulting cDNA was used for quantitative PCR analysis with LightCycler 480 SYBGREEN (Roche Diagnostics, Mannheim, Germany). As an endogenous control, 18S rRNA was used.

#### 2.4.2. Real-Time Quantitative PCR (qPCR)

The qPCR validation experiments of various identified genes were performed in duplicate, and amplification efficiencies were calculated from the standard curve slopes in LightCycler Software 3.3 (Roche Diagnostics, Indianapolis, IN, USA). For the relative quantification of gene expression, the mRNA levels of all genes were normalized to the 18S rRNA levels using the comparative threshold cycle (Ct) method (2^−ΔΔCt^). The results are expressed as fold change relative to the control. Data were obtained from at least three independent experiments and are presented as the mean ± SD.

#### 2.4.3. Western Blot Analysis

After specific treatments, the cells were harvested, and their protein extracts were isolated by cell lysis/extraction reagent according to the manufacturer’s protocol. The protein content was determined using the Bradford method (Bio-Rad, Hercules, CA, USA). Samples with equal amounts of proteins were incubated with the specific primary antibody. Detection was carried out by incubation with secondary horseradish peroxidase-conjugated anti-rabbit or anti-mouse IgG antibody (Santa Cruz Biotechnology, Santa Cruz, CA, USA). The immunoreactive signals were visualized by the chemiluminescent reagent Immobilon Western (Millipore, Billerica, MA, USA) on the X-ray film (GE Healthcare, Piscataway, NJ, USA). The primary antibodies were used according to the selected proteins, including GADD34 (Proteintech; 10449-1-AP), GADD45A (Proteintech; 13747-1-AP), GABARAPL1 (GeneTex; GTX129277), p-eIF2α (Cell signaling; #9721), ATF4 (Cell signaling; #11815), CHOP (Cell signaling; #2895), SirT3 (Cell signaling; #5490), FOXO3A (Cell signaling; #12829), and p-AKT (Cell signaling; #4060). Immunoblot of GAPDH (Abcam; ab128915) was performed to demonstrate equal protein loading.

### 2.5. Statistical Analysis

To analyze the RNA-sequencing results, statistically significant expression changes between MPP^+^ treatment and control in the selected cell line were estimated, and the false discovery rate adjusted *p*-value (q-value) was calculated. Only those differentially expressed genes (DEGs) triggered in response to MPP^+^ treatment with the log2 ratio ≥ 2 and q-value < 0.05 among tested haplogroup were selected for this study. For viability and protein analyses, the result was expressed as mean ± SD. Multiple comparisons were carried out by *t*-test or one-way analysis of variance in SPSS 11.5 for Windows (Chicago, IL, USA). *p*-values < 0.05 were considered significant.

## 3. Results

### 3.1. Viability Test to Select Specific mtDNA Cybrid Lines Exhibiting Resistance to MPP^+^ Treatment

Our previous study of cybrids harboring common Asian mtDNA haplogroups identified B5 as the most resistant to MPP^+^ treatment, whereas D5 was the most vulnerable [9]. In this study, we identified W3 as the most resistant and H11 as the most vulnerable cybrids harbored by European mtDNA haplogroups. These four cybrid lines were then subjected to further investigation to determine levels of resistance/vulnerability to MPP^+^ treatment. As shown in Figure 1, the viability study of these four cybrid lines (B5, D5, H11, and W3) confirmed that the W3 cybrids exhibited the most resistance, whereas H11 exhibited the most sensitivity. Identical results were observed after reproducing the experiments using three different single colonies of each cybrid line.

### 3.2. Transcriptome Analysis to Identify the Genes Linked to MPP^+^ Resistance

#### 3.2.1. Cybrid Sampling and Effects of MPP^+^ on Cybrid Transcriptomes

We hypothesized that the cybrids expressing significantly up-regulated genes could provide increased adaptation or resistance response to external stimulation; therefore, we targeted the W3 cybrid line for further investigation as it was most tolerant to MPP^+^ treatment in our previous viability study. The W3 cybrids were treated with 4 mM MPP^+^, using their corresponding untreated cybrid lines as the control, and then sent for transcriptome profiling using next-generation sequencing as a platform. The criteria for differentially expressed genes (DEGs) included (1) genes that changed by reads per kilobase of transcript per million mapped reads (RPKM) >0.3 and (2) a minimum of twofold difference in normalized read counts between groups. After the profiling, the analysis identified a total of 86 genes in the W3 haplogroup cybrids, which fulfilled the criteria for DEGs between the MPP^+^-treated and -untreated cybrid lines (Appendix A). The *p*-value was estimated for each gene and corrected for multiple testing using the Benjamini–Hochberg correction. The log2 fold change (FC) was used to partition the genes into up- and down-regulated groups [19].

#### 3.2.2. Selection Criteria for Genes Associated with MPP^+^ Resistance

Among the identified DEGs, 51 up- and 35 down-regulated genes were noted after MPP^+^ treatment. The regulated status of each DEG is shown in Appendix A. We subsequently checked the Kyoto Encyclopedia of Genes and Genomes (KEGG) pathways associated with each of the 51 up-regulated genes in the W3 cybrid and interpreted their significance as a potential pathogenetic etiology. A total of 41 of the up-regulated genes were identified as protein-coding gene types. After checking the gene networks involved in the pathogenetic pathway of Parkinson’s disease, we selected six genes, including *PPP1R15A* (protein phosphatase 1 regulatory subunit 15A), *TMEM40* (transmembrane protein 40), *GADD45B* (growth arrest and DNA damage-inducible beta), *GADD45A* (growth arrest and DNA damage-inducible alpha), *DNAJB1* (DnaJ heat shock protein family 40 member B1), and *GABARAPL1* (GABA receptor-associated protein-like 10) for further investigation.

### 3.3. mRNA and Protein Studies to Clarify the Role of Specific Genes in Cybrids Exhibiting Resistance to MPP^+^

We further analyzed the protein expressions of the six selected RNA-related genes identified in the W3 cybrid line after MPP^+^ treatments. Consistent with the RNASeq analysis, the expressions of *PPP1R15A*, *GADD45A*, and *GABARAPL1* increased in the W3 cybrid; however, the increased expressions of the other three identified genes did not achieve statistical significance. The mRNA levels and protein expressions of *PPP1R15A*, *GADD45A*, and *GABARAPL1* were also tested in the D5, B5, and H11 cybrid lines after treatment with MPP^+^. Among these three genes, both the mRNA levels and protein expressions of *PPP1R15A* and *GABARAPL1* exhibited statistically significant increases across all three cybrid lines; although, while we noted statistical increases in the mRNA levels of *GADD45*, the protein expressions did not achieve statistical significance across the three cybrid lines (Appendix A).

### 3.4. GADD34 Protein and the Related mtUPR Pathway Associated with MPP^+^ Resistance

Based on the significantly up-regulated genes identified by our protein studies, and since the *PPP1R15A*-encoding protein GADD34 is a major regulatory component within the mtUPR pathway, we further investigated the role of GADD34 and its associated mtUPR pathway in the pathogenesis of MPP^+^ resistance. We selected the D5, H11, and W3 cybrids for these experiments. Although B5 indeed exhibited resistance to MPP^+^ treatment, we did not include this cybrid line for subsequent investigations as the data did not express significant differences compared to the most resistant haplogroup W3, which could confound the related data analyses. At two hours after MPP^+^ treatment, phosphorylation of eIF2a, which is known to be a signal to initiate mtUPR, was noted in all cybrids (Figure 2). At 6 h after MPP^+^ treatment, significantly elevated expressions of ATF4 and CHOP were observed. Significantly elevated expressions of GADD34 were also noted in all cybrids at 6 h (Figure 3). These results are consistent with previous studies and suggest initiation of the first stage of mtUPR to clear the abnormal/dysfunctional proteins from cells. Of note, expressions of ATF4 and CHOP were significantly lower in the MPP^+^-resistant W3 cybrid than in the D5 and H11 cybrids, while expressions of GADD34 exhibited no significant differences between cybrids. At 24 h after MPP^+^ treatment (Figure 4), although persistently elevated expression of eIF2a phosphorylation was noted, expression of ATF4 was significantly reduced; in addition, expression of CHOP was reduced, although it failed to achieve significance. Collectively, these results could indicate cessation of the initial stage of mtUPR at 24 h. Meanwhile, expressions of GADD34 also remained elevated, which has also been reported in previous studies and indicates initiation of the second stage of mtUPR, which allows cells to recover protein synthesis. Interestingly, we found W3 to exhibit a relatively lower expression of GADD34 as compared to D5 and H11, which may indicate that the W3 cybrid facilitates a relatively prolonged mtUPR process.

### 3.5. Study Results for Other Axes of the Mammalian mtUPR Process

To clarify whether other axes of the mtUPR pathways are involved in the relative resistance to MPP^+^ of cybrids carrying the W3 mtDNA haplogroup, we conducted studies to determine the roles of two additional mtUPR pathways in all three cybrids (Figure 5). At 2 and 6 h after the MPP^+^ treatment, the SIRT3 and phosphate AKT (p-AKT) proteins, which are major regulatory signals for the mtUPR pathways of the sirtuin axis and estrogen receptor α axis, showed no significant changes. In addition, the downstream components of FOXO3A also showed no significant change at 6 and 24 h after the MPP^+^ treatment. However, decreased expressions of SIRT3 and p-AKT were observed at 24 h after the MPP^+^ treatment. The decreases were similar among the cybrid lines, indicating the responses of the sirtuin axis and estrogen receptor α axis are not affected by mtDNA haplogroups.

## 4. Discussion

In this study, we used cellular models harboring various common Asian and Caucasian mtDNA haplogroups to identify the resistant and vulnerable haplotypes under simulated PD conditions. The MPP^+^ treatment revealed haplogroup W3 as the most resistant, while haplogroups D5 and H11 were more vulnerable. The subsequent transcriptome analysis of the resistant W3 cybrids identified several genes exhibiting either up- or down-regulation after MPP^+^ treatment. By analyzing the DEGs, we identified GADD34 for further investigation due to its notably up-regulated expression, which was consistently correlated with the resistant characteristics of W3. In addition, we noted elevated expressions of GADD34 among cybrids harboring all other mtDNA haplogroups, indicating a potential role in the underlying association between certain mtDNA haplogroups and resistance to PD generation. The identification of GADD34 and our subsequent cellular model studies provide evidence of its role in the association between mtDNA and PD. Our study offers a novel method to identify the factors involved in the interaction between the nuclear genome and mtDNA, which could be applied in future investigations of various diseases and their associations with mtDNA variants.

The GADD34 protein, which is encoded by the *PPP1R15A* (protein phosphatase 1 regulatory subunit 15A) gene, is a regulatory component of the canonical axis of mtUPR as it is involved in the phosphatizing and de-phosphatizing processes of eIF2-α [20]. Phosphorylation of eIF2-α by a stress-inducing PKR-like ER-associated kinase (PERK) is a hallmark step in the initiation of mtUPR. It increases the translation of a few selected transcripts, such as activating transcription factor 4 (ATF4) and CCAAT-enhancer-binding protein homologous protein (CHOP), which subsequently trigger the translation of several downstream components involved in mtUPR, including chaperons and proteases for apoptosis [21]. Meanwhile, it attenuates global protein synthesis, allowing cells to clear out misfolded proteins. The recovery of protein synthesis after mtUPR is essential for resuming ordinary cellular functions, wherein GADD34 has been identified as a crucial regulator [22]. Previous studies have found that GADD34 functions through a scaffold complex together with protein phosphatase 1 not only at the initial stage of mtUPR but also at the secondary dephosphorylation stage of eIF2a, thereby allowing cells to recover normal protein synthetic function [23].

In this study, we observed an elevated expression of phosphorylated eIF2-α with subsequent elevated expressions of ATF4 and CHOP, which can initiate the first stage of mtUPR at 2–6 h of simulated injury with MPP^+^ treatment. In the later stage (24 h), persistently elevated expressions of the GADD34 protein, together with a lower expression of ATF4, which signifies a gradual cessation of mtUPR, were noted in all cybrids. However, a relatively lower GADD34 expression in the late stage was noted in the W3 haplogroup, potentially indicating the facilitation of a prolonged mtUPR process and delayed recovery of protein synthesis, which may allow cells to sustain protein homeostasis. It is reasonable to suggest that a more extensive stress-induced injury may require a more prolonged mtUPR process, thereby exerting more phosphorylation. Meanwhile, a higher expression of GADD34 may shorten the mtUPR process, potentially resulting in an incomplete or insufficient clearance of dysfunctional protein and impaired protein homeostasis. Indeed, recent in vitro studies using GADD34 inhibitors as interventional agencies have demonstrated protective effects against the generation of Alzheimer’s and Parkinson’s diseases [24,25].

During its lifespan, a cell will inevitably undergo damage by various external environmental factors and internal oxidative injuries. To ameliorate these risks, cells have developed the ability to remove or repair damaged organelles and proteins. At least three pathways involved in the process of degrading dysfunctional proteins and maintaining protein homeostasis have been identified [14]. In the present study, we investigated the sirtuin axis and estrogen receptor α axis for the relationship between variant mtDNA and resistance of PD generation but found no consistent results. The mitochondrion is vulnerable to cellular metabolic stress, which may be associated with impaired protein folding and import; meanwhile, mtUPR results from retrograde signaling from mitochondria to the nucleus and serves as a protective machinery to restore mitochondrial proteostasis. In terms of the associations between mtDNA and mtUPR investigated here, while we reveal that the W3 haplogroup may prolong the mtUPR process, it has been suggested that the activation of mtUPR may also be dependent on the surrounding mtDNA landscape, indicating multiple factors may be involved in the process [26]. In addition, the initiation of mtUPR via eIF2α phosphorylation could be regulated by nuclear-encoded mitochondrial proteins such as fumarate hydratase 1 and OMA1 zinc metallopeptidase [27,28]. Further investigation is required to elucidate the mechanisms that underlie the reciprocal molecular communications between mitochondria and the nucleus during the process of mtUPR, as well as the potential involvement of mtDNA variants.

There are certain limitations to our present study. Although the use of a uniform nuclear genome in the cybrid model allowed us to explore the influence of mtDNA variants on the development of PD, we did not investigate the impact of different nuclear genomes. Indeed, it is possible that a nuclear genomic variation inherited together with the mitochondrial haplogroup could contribute to PD development. In addition, regarding the transcriptome profiling study, our screen test to identify the genes linked to PD development only involved the most resistant W3 cybrid. Thus, no transcriptome studies of cybrids carrying other mtDNA haplogroups were conducted to reinforce our results or for comparison purposes, thereby limiting the generalizability of the results. Furthermore, although the molecular biological study consistently identified GADD34 as an important component in all cybrid lines, other nuclear-mitochondrial communication pathways require further investigation, while factors that could down-regulate GADD34 to prolong protein homeostasis in the late phase of stress also warrant study in future research.

## 5. Conclusions

In conclusion, our study suggests that mtDNA haplogroup-related PD resistance is associated with the inhibited expression of GADD34 and a subsequently prolonged mtUPR process, which may offer neuroprotective effects. Revealing the exact mechanisms underlying the association between mtUPR and the PD-resistant W3 haplogroup may serve as a preliminary base for the development of a novel neuroprotective strategy for the treatment of PD and other mitochondria-related neurodegenerative disorders.

## Figures and Tables

**Figure 1 cells-13-00694-f001:**
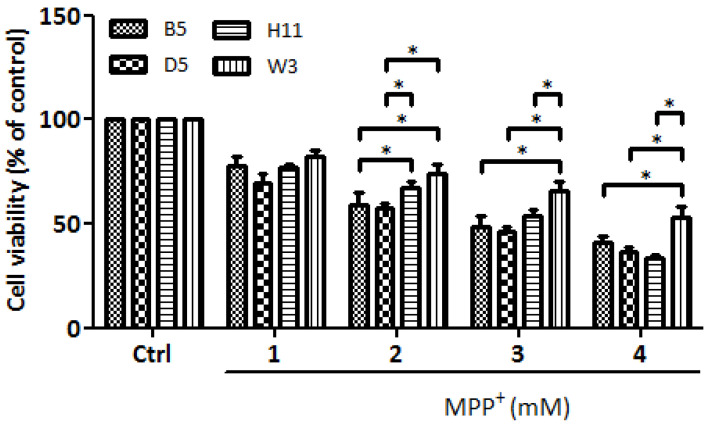
Cytotoxic effects of mitochondrial complex I inhibitor MPP^+^ on B5, D5, H11, and W3 were assessed by WST-1 (Roche) assay. The histogram represents the percentages of viable cells in B5, D5, H11, and W3 cell lines treated with various concentrations of MPP^+^ (1–4 mM) for 24 h. Values are mean ± SD of three independent experiments. A * *p* < 0.05 denotes statistical significance between different cybrid cell lines (one-way ANOVA with Tukey’s post hoc analysis).

**Figure 2 cells-13-00694-f002:**
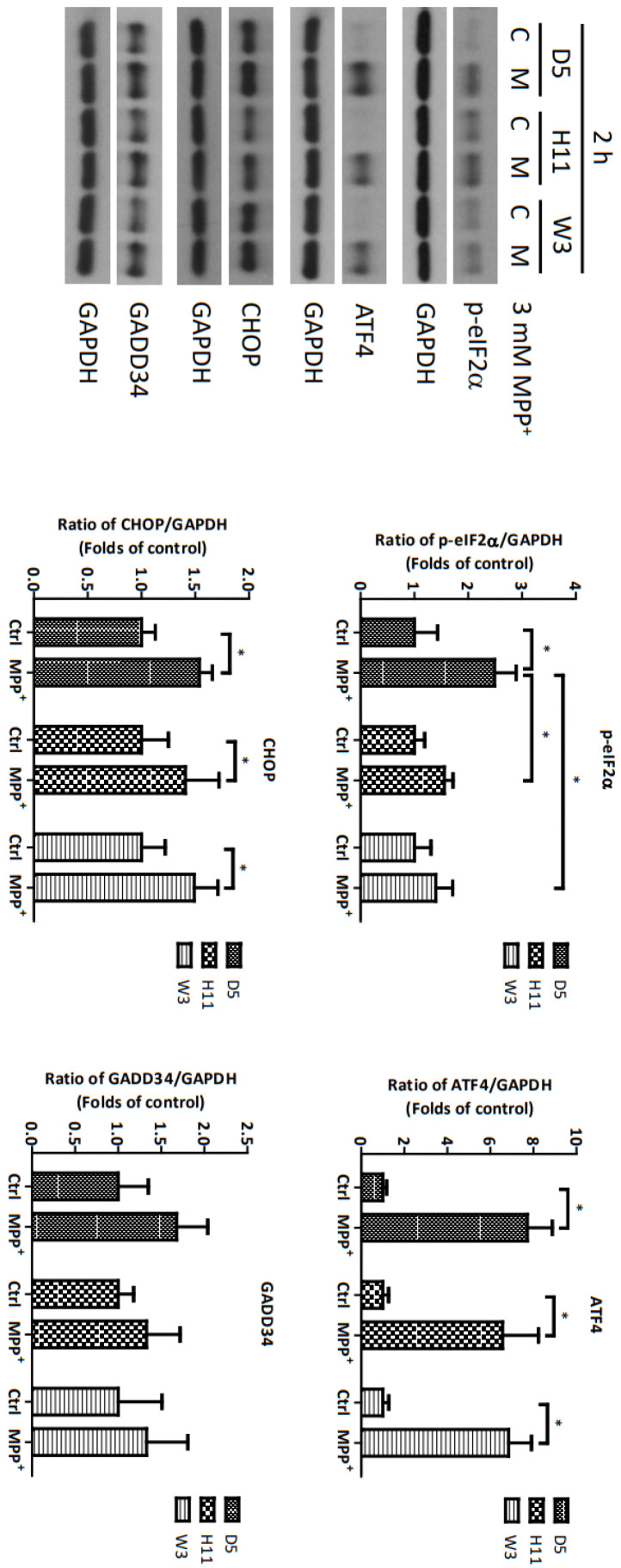
Expressions of p-eIF2α, ATF4, CHOP, and GADD34 at 2 h in D5, H11, and W3 cybrid cells. D5, H11, and W3 cybrid cells were treated with 3 mM MPP^+^ for 2 h. Western blotting analysis of the expressions of p-eIF2α, ATF4, CHOP, and GADD34. GAPDH was an internal loading control. Values are mean ± SD of triplicate. A * *p* < 0.05 compared to the control (one-way ANOVA with Tukey’s post hoc analysis).

**Figure 3 cells-13-00694-f003:**
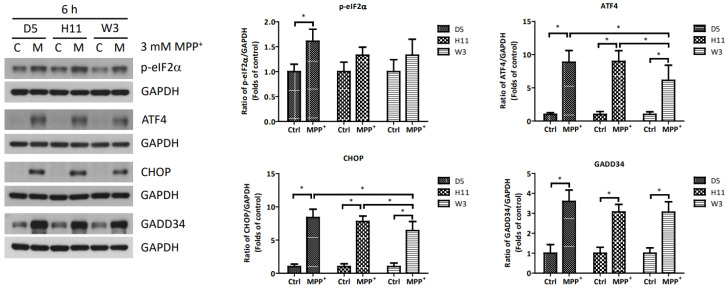
Expressions of p-eIF2α, ATF4, CHOP, and GADD34 at 6 h in D5, H11, and W3 cybrid cells. D5, H11, and W3 cybrid cells were treated with 3 mM MPP^+^ for 6 h. Western blotting analysis of the expressions of p-eIF2α, ATF4, CHOP, and GADD34. GAPDH was an internal loading control. Values are mean ± SD of triplicate. A * *p* < 0.05 compared to the control (one-way ANOVA with Tukey’s post hoc analysis).

**Figure 4 cells-13-00694-f004:**
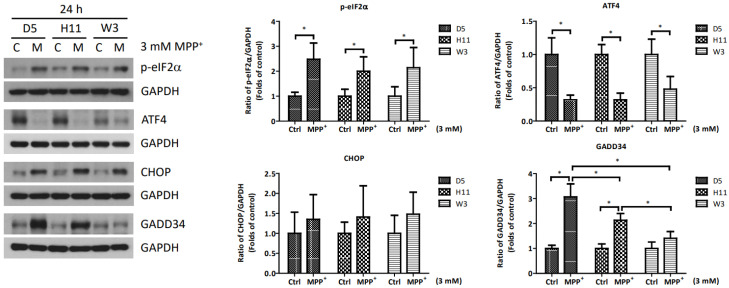
Expressions of p-eIF2α, ATF4, CHOP, and GADD34 at 24 h in D5, H11, and W3 cybrid cells. D5, H11, and W3 cybrid cells were treated with 3 mM MPP^+^ for 24 h. Western blotting analysis of the expressions of p-eIF2α, ATF4, CHOP, and GADD34. GAPDH was an internal loading control. Values are mean ± SD of triplicate. A * *p* < 0.05 compared to the control (one-way ANOVA with Tukey’s post hoc analysis).

**Figure 5 cells-13-00694-f005:**
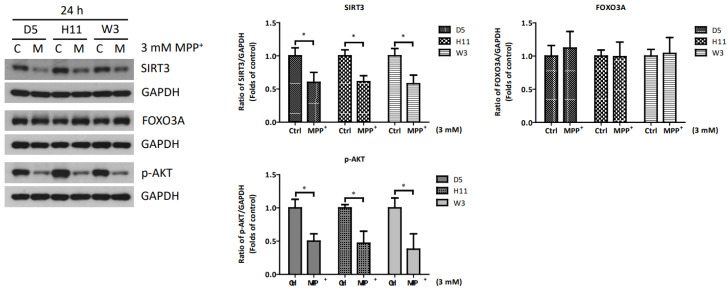
Expressions of SIRT3, FOXO3A, and p-AKT at 24 h in D5, H11, and W3 cybrid cells. D5, H11, and W3 cybrid cells were treated with 3 mM MPP^+^ for 24 h. Western blotting analysis of the expressions of SIRT3, FOXO3A, and p-AKT. GAPDH was as an internal loading control. Values are mean ± SD of triplicate. A * *p* < 0.05 compared to the control (one-way ANOVA with Tukey’s post hoc analysis).

## Data Availability

Data are available from the corresponding author upon reasonable request.

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
