# Peer review of "Unraveling the Pathogenetic Mechanisms Underlying the Association between Specific Mitochondrial DNA Haplogroups and Parkinson’s Disease"

_cells, 2024, doi:10.3390/cells13080694_

Round 1
Reviewer 1 Report
Comments and Suggestions for Authors
The manuscript “Unravelling the Pathogenetic Mechanisms Underlying the Association between Specific Mitochondrial DNA Haplogroups and Parkinson Disease” by Lan et al. investigates the differences in resistance to MPP+ treatment in transmitochondrial cybrids harbouring different mtDNA variants. In this work, Lan and coworkers analyse gene and protein expression profiles in more and less resistant cybrids demonstrating an important role of the mtUPR system in the resistant characteristic of specific mtDNA haplogroups to MPP+ treatment. Finally, the authors demonstrate the implication of GADD34 in generation of PD and propose its potential use as a target in PD’s therapy and prevention.
The experiments well conducted, the results are clearly presented, and English language and style are also fine. The conclusions are supported by the results but I have a couple of comments or questions.
- Why do the authors perform different methods to assay cellular viability in the different cell lines (Alamar blue or WST-1)? I think it should be more comparable if it was evaluated using the same assay for all the samples. On the other hand, in Figure 1 legend, the authors say that the cytotoxic effects were assessed by WST-1 in all cases (In methods section they say that H and W3 cybrids were analyzed using Alamar blue).
- Figure 1: Are differences in cell viability statistically significant?
- Page 6, lines 256-267, the authors talk about differences in gene and protein expression among cell lines, but they don’t show any graph or data.
- Page 6, line 291: authors say that the expression or ATF4 and CHOP were significantly reduced at 24 h after treatment, but no significant differences are observed in CHOP expression in Figure 4.
- When talking about the expression of SIRT3, AKT and FOXO3A, why do authors say that the sirtuin axis and estrogen receptor α axis have no association with the effect of MPP+ on different mtDNA haplogroups if there are significant differences in SIRT3 and p-AKT expression 24 h after the treatment?
- Page 10, lines 414-416: The sentence “In terms of the association between mtDNA and mtUPR…” seems to be incomplete.
- References 6 and 16 are not properly formatted.
Reviewer 2 Report
Comments and Suggestions for Authors
In this research article, the authors investigate the molecular mechanisms linking variants in mitochondrial DNA to Parkinson's disease. For this purpose, they used different cybrid models exposed to MPP+, performed a deeper genomic characterisation in the most resistant line, which presented a specific mtDNA haplotype and an increase in the GADD34-related mitochondrial unfolding protein response, showing a protective role against MPP+ treatment.
The article presents an interesting cell model that is not currently widely used in PD research and an original dataset that is helpful in understanding the role of mtDNA mutations in PD. Unfortunately, the manuscript has serious problems with the reporting of replicates, the analysis of RNA sequencing, and the validation of the relevance of the measured effects.
1) It is not clear from the description of the experiments what the experimental design is. What is meant by two groups of healthy individuals? A description of the number of donors, their age/sex, how samples were selected in the groups (I assume considering different donors) for which experiments would help the reader to understand the experimental design.
2) For all figures, the presentation of individual replicates helps the reader to interpret the variability within samples.
3)The strategy for replication of the data is not clear in the manuscript. How did the authors ensure replication of the experiment (cell batches? cell passages? different experimental days?)?
4) The quality of the manuscript would be significantly improved if the authors could report the variability within replicates.
5) The transcriptomic analysis seems to be underpowered and the analysis is not well documented. From the description of the materials and methods, I understand that the replicates were based on pooling of 3 samples sequenced as 1 sample. This does not allow assessment of replicate heterogeneity, but also does not allow statistical testing. The authors chose to use log (test/control) with a value of 0.001 for the samples that have a value of 0. This approach is particularly problematic when the control has a value of 0 and the power of the test has not been demonstrated to have a sensitivity to detect such low gene expression. Authors should use more sophisticated statistical methods (such as edgeR, ddseq2 or many other equivalent packages) to perform differential gene expression. The reproducibility of the manuscript would be greatly improved if the raw data were made publicly available.Authors report observations on phosphor eiF2alpha and ATF4
Less critical:
6) It is unclear how the RNA was isolated as Trizol is not a kit but a chemical used in cell lysis.
7) Can the authors confirm that the data were really generated on a Solexa sequencer? The platform was released >15 years ago.
8) Were the authors able to find a significant effect of MPP+ and haplotype in Figure 1? If so, what statistical test was used for the analysis?
9)The antibody references used in the study should be reported.
10) How do the MPP+ RNA-Seq data compare with other transcriptomic studies of cells exposed to MPP+?
Reviewer 3 Report
Comments and Suggestions for Authors
The paper titled ‘Unravelling the Pathogenetic Mechanisms Underlying the Association between Specific Mitochondrial DNA Haplogroups and Parkinson Disease’ investigates the association between specific mitochondrial DNA (mtDNA) haplogroups and susceptibility to Parkinson's disease (PD) using cellular models. The introduction contextualizes the study within existing research, highlighting the relevance of mtDNA variants in PD susceptibility and the role of mitochondrial unfolded protein response (mtUPR) in maintaining protein homeostasis. The methods section describes in detail the construction of cybrid cell lines representing different mtDNA haplogroups and the experimental procedures for assessing cellular viability, RNA sequencing, and transcriptome analysis.
Results indicate that cybrids harboring certain mtDNA haplogroups exhibit varying degrees of resistance or vulnerability to MPP+ treatment, a neurotoxin that mimics PD symptoms. Transcriptome analysis identifies genes associated with MPP+ resistance, with a focus on the upregulation of PPP1R15A (GADD34), a key component of the mtUPR pathway. Protein studies confirm elevated expression of GADD34 in resistant cybrids, suggesting its role in mitigating cellular damage.
Discussion of the findings emphasizes the implications for understanding the molecular mechanisms underlying PD susceptibility and highlights the potential of GADD34 as a therapeutic target. The study sheds light on the complex interplay between mtDNA variants, mtUPR pathways, and cellular responses to stress, providing insights that could inform future research and therapeutic interventions for PD and related neurodegenerative diseases.
Overall, the paper provides a comprehensive investigation into the association between specific mitochondrial DNA (mtDNA) haplogroups and Parkinson's disease (PD) susceptibility using cellular models.
I think the idea of this article may be of interest to the readers of Cells. However, some comments, as well as some crucial evidence that should be included to support the authors’ argumentation, needed to be addressed to improve the quality of the manuscript, its adequacy, and its readability prior to the publication in the present form. My overall judgment is to publish this research article after the authors have carefully considered my suggestions below, in particular reshaping parts of the Introduction and Methods sections by adding more evidence.
Please consider the following comments:
• A graphical abstract is highly recommended.
• Abstract: According to the Journal’s guidelines, the abstract should be a total of about 200 words maximum. Please correct the actual one.
• While the introduction effectively outlines the association between specific mitochondrial DNA (mtDNA) haplogroups and susceptibility to Parkinson's disease (PD), it would be valuable to incorporate a brief discussion on the neural substrates underlying PD pathogenesis. Parkinson's disease is characterized by the degeneration of dopaminergic neurons in the substantia nigra pars compacta (SNpc) and the accumulation of Lewy bodies, primarily composed of misfolded alpha-synuclein, in various brain regions. This neuronal loss and protein aggregation lead to disruptions in neurotransmission, particularly within the basal ganglia circuitry, resulting in the motor and non-motor symptoms observed in PD patients. Furthermore, emphasizing the importance of mitochondrial dysfunction in the context of PD neuropathology could enhance the relevance of the study. Mitochondria play a critical role in maintaining cellular energy homeostasis, calcium signaling, and reactive oxygen species (ROS) production, all of which are implicated in PD pathophysiology. Dysfunction in mitochondrial bioenergetics and dynamics has been linked to increased oxidative stress, impaired protein homeostasis, and neuroinflammation, contributing to neuronal vulnerability and degeneration in PD [1-2].
• In this regard, by briefly discussing the neural substrates and mitochondrial dysfunction associated with PD, the introduction can provide a more comprehensive framework for understanding the significance of investigating mtDNA haplogroups in PD susceptibility [3-4]. This contextualization would underscore the relevance of the study's focus on elucidating the molecular mechanisms underlying the observed associations between mtDNA variants and PD pathogenesis.
• I’d recommend providing more context or details about the sample population used for cybrid construction. Information such as age range, gender distribution, and any relevant health characteristics can help contextualize the findings.
• Authors should consider including a brief justification for selecting platelets as the source of mtDNA donors and mitochondria-depleted osteosarcoma cells as the host cells for cybrid construction.
The statistical analysis seems appropriate, with the use of t-tests, ANOVA, and correction for multiple comparisons. However, it would be beneficial to provide more details on the specific statistical tests used for each analysis and how significance was determined. Provide more details on the specific statistical methods used for data analysis, including how data were analyzed for RNA sequencing and protein studies.
• The results section is thorough and presents the findings clearly. However, it would be helpful to include effect sizes or confidence intervals where applicable to provide a better understanding of the magnitude of the observed effects.
• I would ask the authors to discuss any limitations or potential confounding factors that could affect the interpretation of the results.
• The discussion provides a comprehensive interpretation of the results and their implications. It effectively links the findings back to the existing literature and theoretical frameworks. However, it would be beneficial to acknowledge any potential limitations of the study, such as sample size, cell model limitations, or generalizability of findings to human populations.
• Additionally, consider discussing future research directions or practical implications of the findings to further enrich the discussion section.
• The paper generally reads well and maintains a scientific tone. However, there are some instances of repetitive language or overly complex sentences that could be simplified for clarity.
In summary, the paper provides valuable insights into Arc oligomerization, but addressing the above points could enhance the clarity, depth, and overall impact of the findings.
I hope that, after careful revisions, the manuscript can meet the journal’s high standards for publication. I declare no conflict of interest regarding this manuscript.
Best regards,
Reviewer
References:
1. https://doi.org/10.1038/s41398-024-02737-x
2. DOI: 10.3390/ijms242115739
3. DOI: 10.17219/acem/165944
4. https://doi.org/10.3390/ijms25020864
Comments on the Quality of English Language
Minor English editing is required.
Round 2
Reviewer 3 Report
Comments and Suggestions for Authors
Dear Authors,
I am pleased to acknowledge that you have indeed addressed all of my concerns and queries in a clear and precise manner. Your responses have provided valuable insights into the modifications made to the manuscript in light of my comments. It is evident that you have taken great care to ensure that the revised manuscript aligns more closely with the scientific rigor expected for publication in Cells. Having reviewed the revised manuscript, I am satisfied with the changes that have been implemented.
In light of the above, I am pleased to recommend acceptance of your manuscript for publication in Cells.
Best regards,
Reviewer
Academic Editor Notes
The authors should describe the sampling for the transcriptomic analysis also in the results, as it is important.
Response:
Thank you for your suggestion. Actually, we have already mentioned about how to select the W3 cybrid cell line for further transcriptome analysis and the interpretation as well as the selective criteria for the transcriptomic results in the second paragraph of the Results section. The content is shown as below.
However, for allowing readers to read and identify it more clearly, we have changed the title of this paragraph as “ 3.2.1. Cybrid sampling and effects of MPP+ on cybrid transcriptomes” and hope it can meet the requirements.
3.2. Transcriptome analysis to identify the genes linked to MPP+ resistance
3.2.1. Effects of MPP+ on cybrid transcriptomes
We hypothesized that the cybrids expressing significantly up-regulated genes could provide increased adaptation or resistance response to external stimulation; therefore, we targeted the W3 cybrid line for further investigation as it was most tolerant to MPP+ treatment in our previous viability study. The W3 cybrids were treated with 4 mM MPP+, using their corresponding untreated cybrid lines as the control, then sent for transcriptome profiling using next generation sequencing as a platform. -----------
3.2.2. Selection criteria for genes associated with MPP+ resistance
Among the identified DEGs, 51 up- and 35 down-regulated genes were noted after MPP+ treatment. The regulated status of each DEG is shown in Supplementary Table 1. We subsequently checked the Kyoto Encyclopedia of Genes and Genomes (KEGG) pathways associated with each of the 51 up-regulated genes in the W3 cybrid and interpreted their significance as a potential pathogenetic etiology. A total of 41 of the up-regulated genes were identified as protein coding gene types. ------------